# CLIP2UDA: Making Frozen CLIP Reward Unsupervised Domain Adaptation in 3D Semantic Segmentation

Yao Wu
School of Informatics, Xiamen
University
Xiamen, China
wuyao@stu.xmu.edu.cn

Mingwei Xing
Institute of Artificial Intelligence,
Xiamen University
Xiamen, China

Yachao Zhang
Tsinghua Shenzhen International
Graduate School, Tsinghua University
Shenzhen, China

Yuan Xie[†]
East China Normal University
Shanghai, China
Chongqing Institute of East China
Normal University
Chongqing, China

Yanyun Qu[†]
Key Laboratory of Multimedia
Trusted Perception and Efficient
Computing, Ministry of Education of
China, Xiamen University
Xiamen, China
yyqu@xmu.edu.cn

## Abstract

Multi-modal Unsupervised Domain Adaptation (MM-UDA) for large-scale 3D semantic segmentation involves adapting 2D and 3D models to a target domain without labels, which significantly reduces the labor-intensive annotations. Existing MM-UDA methods have often attempted to mitigate the domain discrepancy by aligning features between the source and target data. However, this implementation falls short when applied to image perception due to the susceptibility of images to environmental changes compared to point clouds. To mitigate this limitation, in this work, we explore the potentials of an off-the-shelf Contrastive Language-Image Pre-training (CLIP) model with rich whilst heterogeneous knowledge. To make CLIP task-specific, we propose a top-performing method, dubbed **CLIP2UDA**, which makes frozen CLIP reward unsupervised domain adaptation in 3D semantic segmentation. Specifically, CLIP2UDA alternates between two steps during adaptation: (a) Learning task-specific prompt. 2D features response from the visual encoder are employed to initiate the learning of adaptive text prompt of each domain, and (b) Learning multi-modal domain-invariant representations. These representations interact hierarchically in the shared decoder to obtain unified 2D visual predictions. This enhancement allows for effective alignment between the modality-specific 3D and unified feature space via cross-modal mutual learning. Extensive experimental results demonstrate that our method outperforms state-of-the-art competitors in several widely-recognized adaptation scenarios. Code is available at: https://github.com/Barcaaaa/CLIP2UDA.

†Corresponding authors

## CCS Concepts

• **Computing methodologies → Scene understanding**.

## Keywords

Unsupervised Domain Adaptation, 3D Semantic Segmentation, Multi-modal Learning, Vision-Language Models

### ACM Reference Format:

Yao Wu, Mingwei Xing, Yachao Zhang, Yuan Xie[†], and Yanyun Qu[†]. 2024. CLIP2UDA: Making Frozen CLIP Reward Unsupervised Domain Adaptation in 3D Semantic Segmentation. In *Proceedings of the 32nd ACM International Conference on Multimedia (MM '24), October 28-November 1, 2024, Melbourne, VIC, Australia.* ACM, New York, NY, USA, 10 pages. https://doi.org/10.1145/3664647.3680582

## 1 Introduction

3D scene understanding is the foundation for many real-world applications, such as robotics [33], autonomous driving [40], and augmented reality [19]. Based on the LiDAR point cloud, 3D semantic segmentation is a critical task for scene understanding, which requires assigning semantic labels for each point. However, annotating large-scale datasets for training every new scenario is labor-intensive and time-consuming [38], especially for the tasks demanding point-wise annotations.

Currently, Multi-modal Unsupervised Domain Adaptation (MM-UDA) has been investigated in 3D semantic segmentation, which seeks to adapt 2D and 3D models to a target domain without labels. MM-UDA liberates the annotation of pair-wise 2D and 3D data in the target domain. The core idea of the recent MM-UDA methods [4, 13, 20, 26, 37, 41, 48] usually reinforces the complementation between images and point clouds via mutual learning, to mitigate the discrepancy in data distribution between the source and target domains (See Fig. 1). Albeit successful, these methods usually make the model under-adapted when applied to image perception due to the susceptibility of images to environmental changes compared to point clouds. Particularly with light changes, images lose plentiful context information at night. In this condition, allowing 3D prediction to mimic 2D prediction potentially degrades the performance.

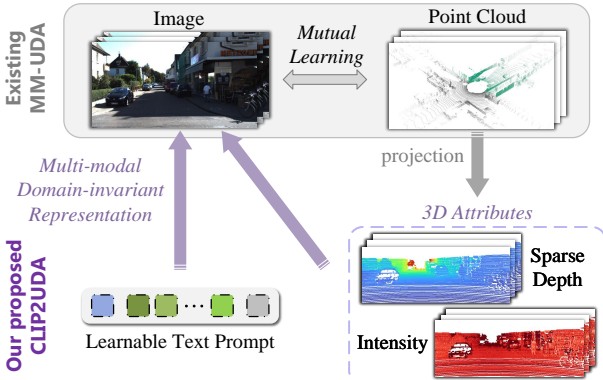

**Figure 1: Unsupervised domain adaptation in 3D semantic segmentation.** *Top*: Existing MM-UDA methods train on paired images and point clouds via mutual learning. *Bottom*: CLIP2UDA incorporates a learnable text prompt derived from the CLIP and inherent 3D attributes derived from the LiDAR sensor, *i.e.,* sparse depth and intensity information, to more robustly adapt 2D and 3D models to the target domain.

Generally, the scarcity of pre-training data hinders training a generalizable 3D model. Vision-Language Models (VLMs) [18, 27, 34] have emerged as the de-facto feature extractor in image classification, captioning, and segmentation nowadays. Such VLMs are trained on massive raw web-curated image-text pairs, achieving promising open-vocabulary recognition. Considering the text description as a domain-invariant prompt, this paper is committed to leveraging rich whilst heterogeneous knowledge from CLIP, allowing us to learn disentangled domain and category representation while avoiding the loss of context information. However, in earlier exploration, simply inheriting the pre-training weights of CLIP and training MM-UDA in a fine-tuning fashion would bring performance degradation. We found it is of utmost importance to avoid any unnecessary attempts to manipulate the visual representations in the space of CLIP. Such a combination may overfit source distribution and potentially lose pre-existing target information.

To this end, two intuitive questions need to be solved: *(i) How to borrow prior knowledge from frozen CLIP?* and *(ii) How to harness prior knowledge to boost 3D semantic segmentation for MM-UDA?* To begin with, for issue *(i)*, considering prompt learning is a popular practice in applying VLMs, we consider designing a task-specific text prompt to be well-tailored for describing domains. Moreover, inspired by MaskCLIP [50], instead of deploying the visual encoder of CLIP as a task network, we serve it as a generalizable auxiliary network without updating its parameters, avoiding introducing additional domain gaps. After that, for issue *(ii)*, we consider introducing inherent 3D attributes derived from LiDAR sensor, *i.e.,* sparse depth and intensity information. Both of them contain spatial structure information and are less influenced by domain shift. By incorporating them in 2D shapes (perspective projection), we aim to learn vision-language-structure correlation to alleviate 3D performance degradation caused by substantial 2D domain shift.

Accordingly, in this paper, we dig deeper into CLIP and propose a top-performing method, dubbed **CLIP2UDA** (See Fig. 1), which

makes frozen CLIP reward unsupervised domain adaptation in 3D semantic segmentation. CLIP2UDA preserves the pre-existing target information from the CLIP model by alternating between two steps during adaptation: (a) Learning task-specific prompts and (b) Learning multi-modal domain-invariant representations. Concretely, for step (a), we introduce Visual-driven Prompt Adaptation (VisPA), using semantic names as the class token to make the data in the source and target domains as close to the label as possible. Meanwhile, to learn the visual distributions in the semantic space and obtain a distribution of prompts per class, VisPA leverages 2D features response from the visual encoder to initiate learning of task-specific text prompts, incorporating domain-specific and image-specific tokens. Nevertheless, relying solely on vision-language correlation is not enough to alleviate the limitation of unstable image perception in environmental changes. Thereupon, for step (b), we propose Text-guided Context Interaction (TexCI), which conducts class-aware fusion and semantic-aware fusion across decoder layers to obtain the text-guided visual feature. This feature is then fused with image and attribute features, and interacts hierarchically in the shared decoder, resulting in unified 2D visual predictions. To summarize, CLIP2UDA tightly associates 3D features with point-wise unified 2D features, enhancing the performing capability of multi-modal domain-invariant representations in 3D scenes that benefit from vision-language-structure correlation.

The main contributions of this paper are as follows:

- We present CLIP2UDA, which exploits prompt learning to transfer the generalization capability of the CLIP model to MM-UDA. It preserves the pre-existing target information from CLIP and learns vision-language-structure correlation.
- VisPA is introduced to leverage 2D features response from the visual encoder to initiate learning of task-specific text prompts. TexCI is introduced to learn multi-modal domain-invariant representation in the shared decoder and obtains unified 2D visual predictions.
- Extensive experimental results demonstrate that our method outperforms state-of-the-art competitors in several widely-recognized adaptation scenarios.

## 2 Related Work

### 2.1 UDA for 3D Semantic Segmentation

UDA methods for 3D semantic segmentation can be grouped as uni-modality cases [16, 31, 39, 43–45, 49] and multi-modality cases [3, 4, 13, 20, 23, 26, 37, 41, 48]. For uni-modality, early methods [39, 49] exploit the generative adversarial network to mitigate domain shift caused by appearance and sparsity differences. Later on, Yuan *et al.* [44, 45] propose the adversarial network based on category-level and prototype-level alignments to address the mismatch of sampling patterns. Complete&Label [43] resolves UDA from the view of the 3D surface completion task. CosMix [31] and ConDA [16] construct an intermediate domain by utilizing joint supervision signals from both the source and target domains for self-training.

Compared to uni-modal cases, multi-modal cases exploit the exclusive information of paired images and point clouds to complement each other. xMUDA [13] first provides a cross-modal mutual learning method for 3D semantic segmentation in UDA. To facilitate learning domain-robust dependencies, several multi-modal

methods extend 2D techniques to learn the domain-invariant representations, such as adversarial learning [23, 26, 48], contrastive learning [41], and style transfer [20]. Based on these dependencies, SSE [48] presents a self-supervised exclusive learning mechanism from plane-to-spatial and discrete-to-textured. BFtD [37] employs cross-modal fusion representation to address imbalanced modality adaptability. Differently, we focus on how to leverage rich whilst heterogeneous knowledge from CLIP [27] to alleviate the 3D performance degradation caused by substantial 2D domain shifts.

## 2.2 Prompt Learning

Vision-Language Models (VLMs) integrate visual and textual inputs to achieve a more comprehensive understanding of the real world, leading to better performance in various tasks. Some of the popular VLMs are CLIP-based models [14, 51, 52] for classification. Recently, several studies have explored the use of prompt learning for dense predictions, which are for the topic of open-vocabulary segmentation in an annotation-free manner [24, 50], supervised segmentation [17, 28] and weakly-supervised segmentation [22, 42].

For 3D scene understanding, there is limited research dedicated to employing prompt learning. Several works [7, 12, 46, 47] align 3D space with open-world language representation, facilitating zero-shot transfer in indoor and outdoor scenes. CLIP2Scene [6] first explores how to establish connections between point and text through self-supervised pre-training, which benefits fine-tuning for 3D downstream tasks. Chen *et al.* [5] leverage the strengths of CLIP and SAM [15] to supervise 2D and 3D networks simultaneously by introducing cross-modality noisy supervision.

Nevertheless, none of the works explore CLIP for UDA in 3D semantic segmentation. To the best of our knowledge, our method is groundbreaking in exploiting CLIP to learn domain-invariant representation for MM-UDA.

## 3 Approach

## 3.1 Preliminary

*3.1.1 Problem Definition.* Given a source domain $\mathcal{D}_S = \{(X_i^{2D,S}, X_i^{3D,S}, Y_i^{3D,S})\}_{i=1}^{n_s}$ with $n_s$ unlabeled 2D images and labeled 3D point clouds, and a target domain $\mathcal{D}_T = \{(X_i^{2D,T}, X_i^{3D,T})\}_{i=1}^{n_t}$ with $n_t$ unlabeled 2D images and 3D point clouds under the condition that the source and target data distributions are not equal. The source and target domains share the same label space and only the source point cloud has annotation $Y^{3D,S}$ belonging to $C$ classes for each 3D point. The task is to learn a model $f : X^{2D,T}, X^{3D,T} \rightarrow Y^{3D,T}$ that could predict the target 3D labels.

*3.1.2 Revisiting CoOp.* Generally, CoOp [52] consists of a visual encoder $\mathcal{F}_V(\cdot)$ and a text encoder $\mathcal{F}_T(\cdot)$ based on CLIP, both jointly trained to map the image and text into a unified representation space. The learnable prompt given to $\mathcal{F}_T(\cdot)$ is designed with the following form:

$$t = [V]_1[V]_2...[V]_M[CLS], \tag{1}$$

where each $[V]_m$ ($m \in \{1, ..., M\}$) is a learnable vector with the same dimension $D$ as textual embedding, $M$ is a hyperparameter that specifies the length of context tokens, and $[CLS]$ is the class

token, which is converted into a low-cased byte pair encoding representation [32]. After that, textual embedding can be represented as $e_t = \mathcal{F}_T(t)$, where $e_t \in \mathbb{R}^{C \times D}$ is the embedding for $C$ classes.

Meanwhile, take ResNet [11] for example, there are 4 stages in total and we denote the feature maps as $\{f_i\}_{i=1}^4$. Different from the original ResNet, a multi-head self-attention (MHSA) layer is performed to concatenate visual features $[\bar{f}_4, f_4]$ of the last layer, where $\bar{f}_4$ is the global features after global average pooling of $f_4$. After that, the global and local visual embeddings can be represented as $[z_g, z_l] = \text{MHSA}([\bar{f}_4, f_4])$, where $z_g \in \mathbb{R}^{1 \times D}$ and $z_l \in \mathbb{R}^{H_4 W_4 \times D}$, $H_4, W_4$ are the height and width of the feature map from the 4-th stage of the backbone, respectively.

## 3.2 Overview

The overall framework of CLIP2UDA is depicted in Fig. 2, which training alternately between two steps during adaptation: *Visual-driven Prompt Adaptation* (VisPA) and *Text-guided Context Interaction* (TexCI). Firstly, in VisPA, images $X^{2D}$ are input to CLIP visual encoder $\mathcal{F}_V(\cdot)$ to initiate learning of a task-specific text prompt, and then generate visual-driven text embedding via CLIP text encoder $\mathcal{F}_T(\cdot)$ and Transformer decoder $\mathcal{G}_T(\cdot, \cdot)$. For the unified branch, the 2D segmentation task involves three types of visual inputs: image $X^{2D}$, sparse depth $X^{Dep}$, and intensity $X^{Int}$. The former input is fed into the 2D encoder $\mathcal{F}_I(\cdot)$. In contrast, the latter two inputs $X^{Dep}$ and $X^{Int}$ are fed into the attribute encoder $\mathcal{F}_A(\cdot)$ for acquiring a compact representation of 3D attributes in 2D space. Afterward, TexCI is introduced to learn multi-modal domain-invariant representation in the shared decoder $\mathcal{G}_I(\cdot, \cdot, \cdot)$ and obtains point-wise unified features. Finally, after extracting 3D features output from the 3D network $\mathcal{G}_P(\mathcal{F}_P(\cdot))$, we can bridge the cross-domain gaps by mutually learning between unified predictions and 3D predictions.

## 3.3 Visual-driven Prompt Adaptation

Improving sentence structure through visual descriptions of natural objects helps reduce the disparity between textual and visual cues. Additionally, this principle holds in the context of autonomous driving scenarios. For instance, "*A yellow car driving in the night scene.*" is more exact. However, incorporating rich text into the manual prompts is time-consuming, as it has to be based on trial and error, and does not guarantee an optimal prompt either. In this regard, VisPA is introduced to learn prompts for each class directly from the 2D visual domain to effectively encode the visual distribution, as opposed to the static prompting technique. To this end, we denote the prompt for one class with the class token $[CLS]$ given an image $X^{2D}$ as:

$$t_{vis} = [D]_{S/T}[\tilde{V}]_1[\tilde{V}]_2 \cdots [\tilde{V}]_M[CLS], \tag{2}$$

where $[D]_{S/T}$ and $[\tilde{V}]_m$ denote the Domain-specific Token (D-Token) and the Image-specific Token (I-Token), respectively. Both of them are dedicated to capturing visual context using a parameter-efficient design, to boost the adaptability of text prompts.

*3.3.1 D-Token.* To capture domain-specific information, we sample a feature vector from the semantic distribution of multi-layer features as domain-specific vectors and subsequently map it onto $[D]_{S/T}$. The first and second-order batch-wise feature statistics are considered indicators of domain style: $[\mu, \rho]$. Then, a lightweight

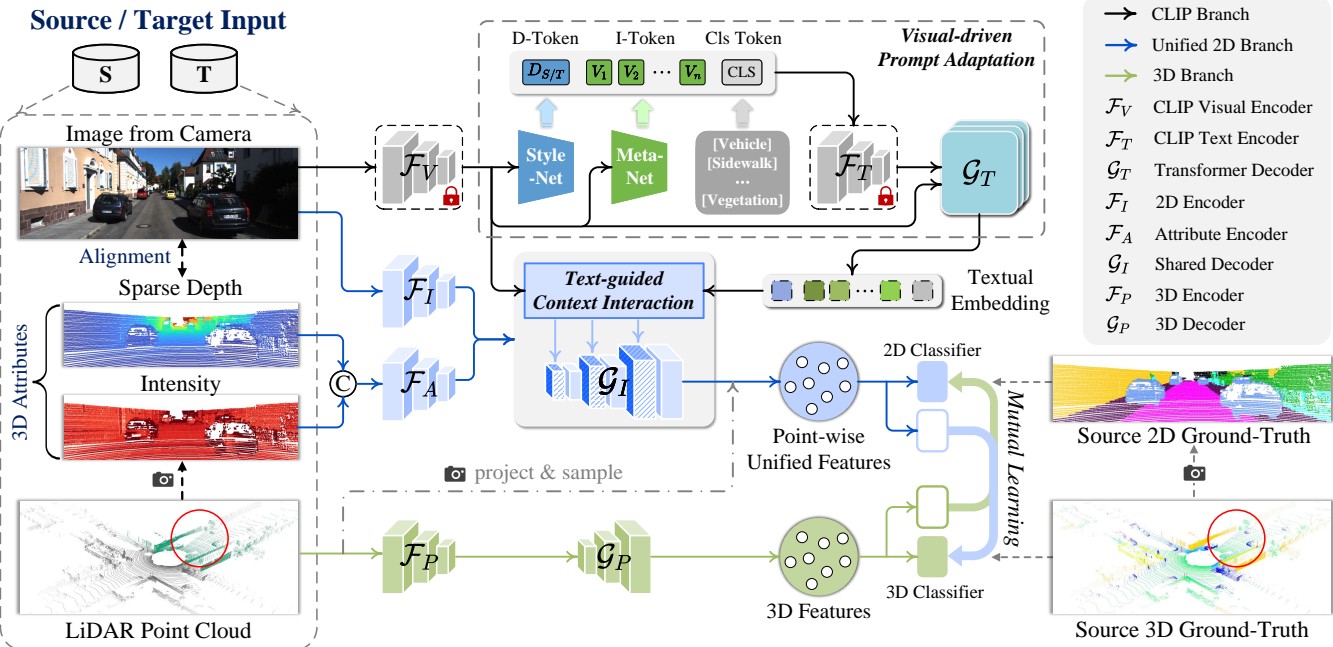

**Figure 2: The framework of CLIP2UDA. CLIP serves as a generalizable auxiliary network without updating its parameters.**

neural network, called Style-Net $h_\phi$ parameterized by $\phi$, calculates $\{(\mu_i, \rho_i)\}_{i=1}^4$ from four layers of $\mathcal{F}_V$. In this work, $h_\phi$ is built with two linear layers corresponding to the output of an encoder layer that reduces the input dimension to the same as textual embedding.

*3.3.2 I-Token.* On top of the learnable vectors $[V]_m$, we further learn another lightweight neural network, called Meta-Net, to generate a conditional token for each input, which is then combined with the context prompt. In this work, the Meta-Net is built with a two-layer bottleneck structure (Linear-ReLU-Linear), with the hidden layer reducing the input dimension by 16×. Let $h_\theta(\cdot)$ denote the Meta-Net parameterized by $\theta$, each image-specific token is obtained by: $[\tilde{V}]_m \leftarrow [V]_m + \gamma$, where $\gamma = h_\theta(z_g)$.

*3.3.3 Visual-driven Textual Embedding.* After utilizing the visual distribution of domain to prompt text tokens, it is crucial to explicitly leverage visual features to drive textual embedding for obtaining more compact representations. Since the attention mechanism can capture the long-range dependency by using pair-wise affinities across all positions, we employ a Transformer decoder $\mathcal{G}_T$ [35], which contains $N$ (=3 in our case) MHSA layers and multi-head cross-attention layers. $\mathcal{G}_T$ builds the interaction between visual embedding $e_v = z_g \uplus z_l$ and textual embedding $e_t = \mathcal{F}_T(t_{vis})$, and then generates refined textual embedding, where $\uplus$ is the concatenate operation. This implementation is written as:

$$\tilde{e}_t = \mathcal{G}_T(e_t, e_v), \tag{3}$$

where $\tilde{e}_t$ encourages textual embedding to mine the most correlated visual cues. Finally, we can obtain visual-driven textual embedding via a skip residual connection:

$$e_{v \rightarrow t} = e_t + \delta \tilde{e}_t, \tag{4}$$

where $\delta$ is a learnable parameter, which is initialized with a small value (*i.e.*, $10^{-4}$) to maximally preserve the language priors from the textual embedding.

## 3.4 Text-guided Context Interaction

Failing the fine-tuning attempt with CLIP pre-trained weights, we turn to a solution that avoids overfit source distribution and potentially loses pre-existing target information. To this end, we relax from this constraint and benefit from more flexible architectures tailored for multi-modal semantic segmentation. That is, we employ the CLIP-based model as an auxiliary network, whose pre-trained parameters are frozen, and only update the learnable prompts $t_{vis}$ in VisPA. Specifically, as depicted in Fig. 3, we devise TexCI for hierarchical multi-modal fusion in the shared decoder, which conducts class-aware fusion (C-Fus) and semantic-aware fusion (S-Fus) across decoder layers.

*3.4.1 C-Fus.* Visual embeddings encompass global semantics about the entire image. We consider that they also carry the dense semantics of multiple objects, as the features of each pixel aggregate information from all other pixels in forwarding. Ideally, we hope to extract local features from CLIP to tightly associate with objects in a 3D scene. Specifically, the visual embedding $e_v$ is leveraged to compute the pixel-text matching map with visual-driven textual embedding $e_{v \rightarrow t}$, which is defined as follows:

$$S = \hat{e}_v \hat{e}_{v \rightarrow t}^\top, \quad S \in \mathbb{R}^{H_4 W_4 \times C}, \tag{5}$$

where $\hat{e}_v$ and $\hat{e}_{v \rightarrow t}$ are the $l_2$ normalized version of $e_v$ and $e_{v \rightarrow t}$ along the channel dimension.

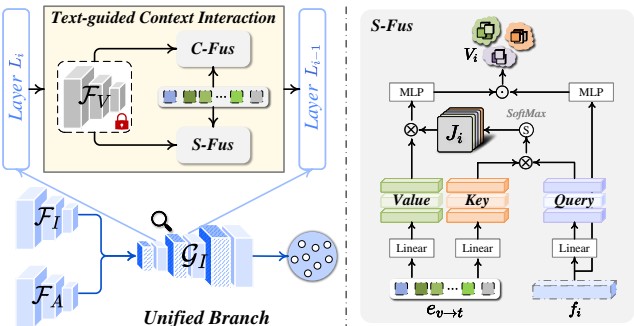

**Figure 3: Architecture of Text-guided Context Interaction in the shared decoder $\mathcal{G}_I$. Each layer $L_i$ implements deconvolution with image feature $f_i^{2D}$ and attribute feature $f_i^{Att}$.**

*3.4.2 S-Fus.* To further explicitly mine the complementary peculiarities of multi-modality, it is essential to align the pixel and point representations of the scene across layers. Given the input $\{f_i\}_{i=1}^4$ and $e_{v\to t}$, S-Fus performs hierarchical multi-modal fusion by using visual embedding $f_i$ as the *Query* and textual embedding $e_{v\to t}$ as the *Key* and *Value*.

Firstly, at each spatial location, S-Fus aggregates $e_{v\to t}$ across the text dimension to generate a joint-modal affinity matrix $J_i \in \mathbb{R}^{C\times H_iW_i}$ via dot-product, which collects textual information most relevant to pixels of the current layer.

$$J_i = SoftMax\left(\frac{\omega_k e_{v\to t} \otimes \omega_q f_i^{\top}}{\sqrt{D_i}}\right). \tag{6}$$

where $\otimes$ means matrix multiplication. Secondly, leveraging the affinity matrix $J_i$, we can preliminarily estimate a set of text-guided visual feature $V_i \in \mathbb{R}^{H_iW_i\times D_i}$ of each layer using the equation:

$$V_i = \omega_o \left(J_i^{\top} \omega_v e_{v\to t}\right) \odot \omega_w f_i, \tag{7}$$

where $\odot$ means element-wise multiplication, $\omega_q$, $\omega_k$, $\omega_v$, $\omega_w$, and $\omega_o$ are linear functions with trainable parameters.

*3.4.3 Shared Visual Decoding.* After that, we adopt the UNet-like decoder [29] as the shared decoder $\mathcal{G}_I$ in the unified branch. Due to the sparsity of LiDAR point clouds, in multi-modal interaction between points and pixels, low-resolution matching maps cannot establish effective one-to-one relationships with corresponding point labels. Hence, considering that the matching map retains the available vision-language prior knowledge, we simply concatenate upsampled $S_i$ and $V_i$ to the $i$-th image feature $f_i^{2D}$ and attribute feature map $f_i^{Att}$ output from $i$-th layers of $\mathcal{F}_I$ and $\mathcal{F}_A$ respectively, explicitly incorporating language prior:

$$f_i^{uni} = \left[f_i^{2D} \uplus f_i^{Att} \uplus S_i \uplus V_i\right], \tag{8}$$

where $f_i^{uni}$ characterizes the multi-scale fusion feature, joint embedding during visual decoding. Consequently, by jointly learning multi-modal domain-invariant representations in $\mathcal{G}_I$, a point-wise unified prediction $\mathbf{P}^{Uni}$ can be defined to handle task and categories characteristics via the 2D classifier.

## 3.5 Unified Cross-modal Learning

The point-wise supervised segmentation loss of the source domain is formulated as follows:

$$\mathcal{L}_{seg} = -\frac{1}{N\times C}\sum_{n=1}^{N}\sum_{c=1}^{C} Y_{(n,c)}^{3D,S} \log \mathbf{P}_{(n,c)}^S, \tag{9}$$

where main prediction $\mathbf{P}^S$ is either $\mathbf{P}^{Uni,S}$ or $\mathbf{P}^{3D,S}$.

The goal of unsupervised learning across modalities is twofold. Firstly, we want to transfer knowledge from the unified modality to the 3D modality on the target-domain dataset. Secondly, we devise mutual learning on source and target domains, where the task is to estimate the prediction of other modalities. Same to xMUDA [13], we choose the Kullback-Leibler divergence $D_{KL}(\cdot||\cdot)$ for the cross-modal loss $\mathcal{L}_{xM}$ and define it as follows:

$$\mathcal{L}_{xM} = D_{KL}(\mathbf{P}^{Uni}||\mathbf{P}^{3D\mapsto Uni}) + D_{KL}(\mathbf{P}^{3D}||\mathbf{P}^{Uni\mapsto 3D}), \tag{10}$$

where $\mathbf{P}^{Uni}$ and $\mathbf{P}^{3D}$ is to be estimated by the mimicking $\mathbf{P}^{3D\mapsto Uni}$ and $\mathbf{P}^{Uni\mapsto 3D}$ from auxiliary classifier, respectively. Note that for convenience, we disregard the domain notation $S/T$ of superscript in prediction. Finally, the overall loss function is defined as:

$$\begin{aligned} \mathcal{L}_{total} = \mathcal{L}_{seg} &+ \lambda_S \mathcal{L}_{xM}(X^{2D,S}, X^{3D,S}, X^{Dep,S}, X^{Int,S}) + \\ &\lambda_T \mathcal{L}_{xM}(X^{2D,T}, X^{3D,T}, X^{Dep,T}, X^{Int,T}), \end{aligned} \tag{11}$$

where $\lambda_S$ and $\lambda_T$ are weights trading off $\mathcal{L}_{xM}$ on source domain and target domain inputs, respectively.

## 4 Experiments

### 4.1 Datasets

For evaluation, we use four public autonomous driving datasets, including three real scenarios: *nuScenes* [2], *SemanticKITTI* [1], *A2D2* [9] and one synthetic scenario: *VirtualKITTI* [8]. For all real datasets, LiDAR and RGB cameras are synchronized and calibrated, allowing 2D-to-3D projection, and for the synthetic dataset, *VirtualKITTI* provides depth maps so we simulate LiDAR scanning via uniform point sampling. Furthermore, following [13], we only use the front camera image and the corresponding LiDAR points.

Our experimental scenarios cover typical real-to-real domain adaptation challenges like lighting changes (*nuScenes*: *Day → Night*), scene layout of country (nuScenes: *USA → Singapore*), and sensor setups (*A2D2 → SemanticKITTI*). For the first two scenarios, we choose 6 merged classes while for the last scenario, we select 10 shared classes between two datasets. In addition, the synthetic-to-real domain adaptation challenge also be considered (*VirtualKITTI→SemanticKITTI*, simulated depth, and RGB to real LiDAR and camera, with 6 merged classes). Details are provided in supplementary materials.

### 4.2 Implementation Details

For the 2D backbone, we use a modified version of U-Net [29], which consists of a dual-branch ResNet-34 [11] as the 2D encoder and Attribute encoder and transposed convolutions with skip connections as the Shared decoder. It is worth noting that ResNet for the 2D encoder is pre-trained on ImageNet [30], while the Attribute encoder is trained from scratch. Meanwhile, depth and intensity information are 3D attributes derived from LiDAR sensors. Therefore,

**Table 1: Quantitative results (mIoU, %) on four settings. The best value of "2D+3D" is marked in red, and the second best value is marked in blue. "†" indicates the reproduced results. "$_{PL}$" denotes retraining with pseudo labels.**

| Info | Method | nuScenes: Day→Night | | | nuScenes: USA→Sing. | | | Virt.KITTI→Sem.KITTI | | | A2D2→Sem.KITTI | | |
|---|---|---|---|---|---|---|---|---|---|---|---|---|---|
| | | 2D | 3D | **2D+3D** | 2D | 3D | **2D+3D** | 2D | 3D | **2D+3D** | 2D | 3D | **2D+3D** |
| Lower bound | Source-only | 47.8 | 68.8 | 63.3 | 58.4 | 62.8 | 68.2 | 26.8 | 42.0 | 42.2 | 34.2 | 35.9 | 40.4 |
| Uni-modal | Deep logCORAL [25] | 47.7 | 68.7 | 63.7 | 64.4 | 63.2 | 69.4 | 41.4 | 36.8 | 47.0 | 35.1 | 41.0 | 42.2 |
| | MinEnt [36] | 47.1 | 68.8 | 63.6 | 57.6 | 61.5 | 66.0 | 39.2 | 43.3 | 47.1 | 37.8 | 39.6 | 42.6 |
| | BDL$_{PL}$ [21] | 47.0 | 69.6 | 63.0 | 62.0 | 64.8 | 70.4 | 21.5 | 44.3 | 35.6 | 34.7 | 41.7 | 45.2 |
| Multi-modal | xMUDA [13] | 55.5 | 69.2 | 67.4 | 64.4 | 63.2 | 69.4 | 42.1 | 46.7 | 48.2 | 38.3 | 46.0 | 44.0 |
| | AUDA† [23] | 55.6 | 69.8 | 64.8 | 64.0 | 64.0 | 69.2 | 35.8 | 37.8 | 41.3 | 43.0 | 43.6 | 46.8 |
| | DsCML† [26] | 50.9 | 49.3 | 53.2 | 65.6 | 56.2 | 66.1 | 38.4 | 38.4 | 45.5 | 39.6 | 45.1 | 44.5 |
| | Dual-Cross† [20] | 58.5 | 69.7 | 68.0 | 64.7 | 58.1 | 66.5 | 40.7 | 35.1 | 44.2 | 44.3 | 46.1 | 48.6 |
| | SSE-xMUDA† [48] | 62.8 | 69.0 | 68.9 | 64.9 | 63.9 | 69.2 | 45.9 | 40.0 | 49.6 | 44.5 | 46.8 | 48.4 |
| | BFtD-xMUDA† [37] | 57.1 | 70.4 | 68.3 | 63.7 | 62.2 | 69.4 | 41.5 | 45.5 | 51.5 | 40.5 | 44.4 | 48.7 |
| | **CLIP2UDA (Ours)** | 73.1 | 71.5 | 74.1 | 71.6 | 68.3 | 74.0 | 57.8 | 53.0 | 60.4 | 45.4 | 45.5 | 50.0 |
| | △$_{Gain}$ | ↑ 10.3 | ↑ 1.1 | ↑ 5.8 | ↑ 6.7 | ↑ 4.4 | ↑ 4.6 | ↑ 11.9 | ↑ 6.3 | ↑ 8.9 | ↑ 0.9 | ↓ 1.3 | ↑ 1.3 |
| | xMUDA$_{PL}$ [13] | 57.6 | 69.6 | 64.4 | 67.0 | 65.4 | 71.2 | 45.8 | 51.4 | 52.0 | 41.2 | 49.8 | 47.5 |
| | AUDA$_{PL}$† [23] | 54.3 | 69.6 | 61.1 | 65.9 | 65.3 | 70.6 | 35.9 | 45.5 | 45.9 | 46.8 | 48.1 | 50.6 |
| | DsCML$_{PL}$† [26] | 51.4 | 49.8 | 53.8 | 65.6 | 57.5 | 66.9 | 39.6 | 41.8 | 42.2 | 46.8 | 51.8 | 52.4 |
| | Dual-Cross$_{PL}$† [20] | 59.1 | 69.0 | 68.2 | 66.5 | 59.8 | 68.8 | 43.1 | 39.4 | 47.6 | 44.9 | 52.8 | 52.3 |
| | SSE-xMUDA$_{PL}$† [48] | 59.1 | 67.0 | 66.3 | 66.9 | 64.4 | 70.6 | 47.2 | 53.5 | 55.2 | 45.9 | 51.5 | 52.5 |
| | BFtD-xMUDA$_{PL}$† [37] | 60.6 | 70.0 | 66.6 | 65.9 | 66.0 | 71.3 | 48.6 | 55.4 | 57.5 | 42.6 | 53.7 | 52.7 |
| | **CLIP2UDA$_{PL}$ (Ours)** | 73.3 | 71.6 | 74.2 | 74.8 | 69.9 | 75.8 | 59.6 | 55.4 | 62.9 | 45.4 | 50.4 | 52.7 |
| | △$_{Gain}$ | ↑ 12.7 | ↑ 1.6 | ↑ 7.6 | ↑ 7.8 | ↑ 3.9 | ↑ 4.5 | ↑ 11.0 | 0.0 | ↑ 5.4 | ↑ 2.8 | ↓ 3.3 | 0.0 |
| Upper bound | Target-only | 61.5 | 69.8 | 69.2 | 75.4 | 76.0 | 79.6 | 66.3 | 78.4 | 80.1 | 59.3 | 71.9 | 73.6 |

our method does not introduce extra datasets to ensure fairness in equal experimental conditions. For the 3D backbone, we use the official SparseConvNet [10] implementation. The voxel size is set to 5cm which is small enough to only have one 3D point per voxel.

For the visual encoder of CLIP, all main experiments adopt CLIP-pretrained ResNet-50, while the text encoder of CLIP is built on top of a Transformer [35]. Of note, we fix the parameters of text and visual encoders during training to preserve the natural language knowledge learned from large-scale pre-training. To reduce the computational costs, we project both the 2D visual embeddings and the textual embeddings to a lower dimension (256) before feeding into the Transformer module.

We employ standard 2D/3D data augmentation and log-smoothed class weights on point-wise supervised segmentation loss to address the class imbalance. The batch size is set to 8. Our model is trained on real-to-real adaptation for 100k iterations. We utilize an iteration-based learning schedule where the initial learning rate is 0.001 and then it is divided by 10 at 80k and 90k iterations. For synthetic-to-real, the training is performed for 30k iterations, and the learning rate is divided by 10 at the 25k and 28k iterations. As regards the hyper-parameters, $\lambda_S$ and $\lambda_T$ in cross-modal loss are set to 1.0 and 0.1 on "Day→Night" and "USA→Sing.", 0.1 and 0.02 on "Virt.KITTI→Sem.KITTI" and "A2D2→Sem.KITTI" respectively, without performing any fine-tuning on these values. All experiments are conducted on a single NVIDIA RTX 3090 GPU.

### 4.3 Quantitative and Qualitative Comparison

We compare our method with three typical 2D UDA methods, which can be easily extended to multi-modal UDA tasks. Moreover, six multi-modal UDA methods are discussed and they are roughly divided into two types: bridging the cross-modality gap, such as xMUDA [13] and BFtD [37]; and bridging the cross-domain gap, such as AUDA [23], DsCML [26], Dual-Cross [20], and SSE [48]. The comparison results for 3D semantic segmentation in mean Intersection over Union (mIoU) on the target testing data are shown in Tab. 1. Overall, CLIP2UDA achieves the best performance in all scenarios against the competitors, $w.r.t.$ ensemble result "2D+3D".

The source-only model is the lower bound, which is not domain adaptation as it is only trained on the source-domain dataset. It is observed that our method brings a significant adaptation effect on all scenarios compared to the source-only model, with the gains of 10.8%, 5.8%, 18.2%, and 9.6% in mIoU, respectively. Compared with the baseline (xMUDA), our method exceeds it by large margins with gains of 6.7%, 4.6%, 12.2%, and 6.0% in mIoU. Compared to the best results in all multi-modal learning methods, it is observed that our method achieves 5.8%, 4.6%, 8.9%, 1.3% mIoU gains over all scenarios. Of note, our method typically yields a higher score, up to +1∼12% mIoU on each 2D/3D separate branch (See △$_{Gain}$). Especially in synthetic-to-real scenarios, it has been proven that point clouds with visual language priors are beneficial for addressing the adaptation problem in domains with significant discrepancy. Furthermore, cross-modal learning and self-training with pseudo-labels (PL) are complementary in their combination. When re-trained with PL, our model still achieves superior performance.

In Fig. 4, we visualize the segmentation results of four settings. Compared to xMUDA and BFtD, CLIP2UDA avoids erroneous and mingled predictions in many small (person) and large (sidewalk, vegetation) regions, showing versatility across all scenarios.

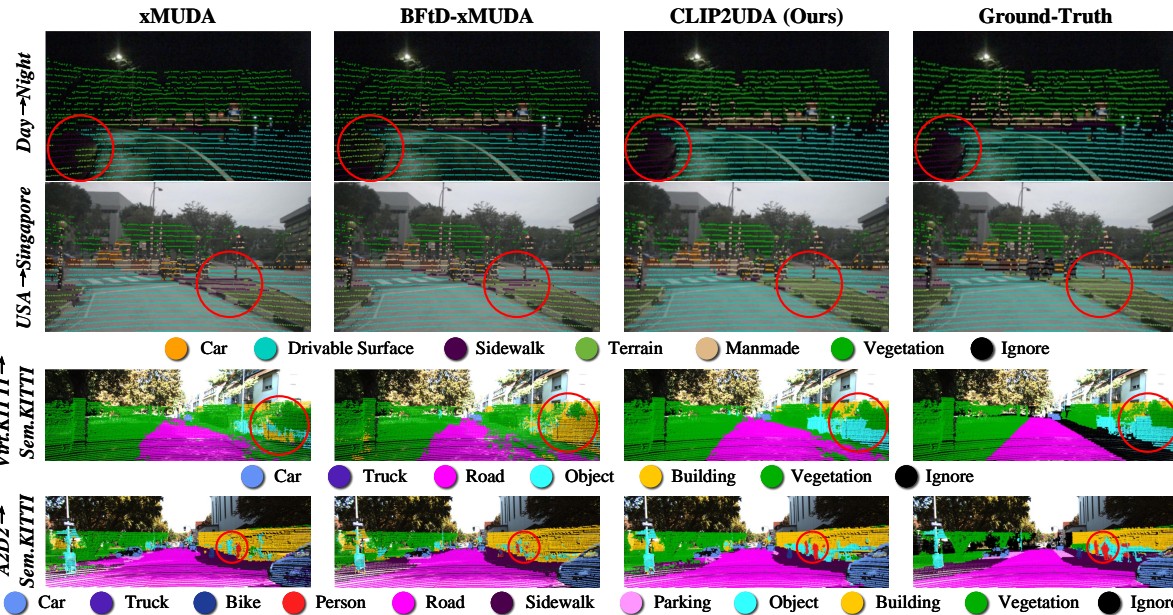

**Figure 4: Qualitative results. We showcase the ensembling result "2D+3D" on the target test set of four adaptation scenarios. The differences are highlighted with red circles.**

**Table 2: Ablation study on the effectiveness of significant components in CLIP2UDA. Attr.: Apply 3D attributes.**

|    | Attr. | VisPA | TexCI | | USA→Sing. | | | V.K.→S.K. | | |
|----|-------|-------|-------|-------|------|------|--------|------|------|--------|
|    |       |       | C-Fus | S-Fus | 2D   | 3D   | 2D+3D  | 2D   | 3D   | 2D+3D  |
| xMUDA | | | | | 64.4 | 63.2 | 69.4 | 42.1 | 46.7 | 48.2 |
| #1 | ✓ |   |   |   | 70.7 | 67.2 | 72.8 | 54.7 | 48.0 | 56.4 |
| #2 | ✓ | ✓ | ✓ |   | 71.1 | 68.2 | 73.2 | 56.3 | 48.8 | 58.1 |
| #3 | ✓ | ✓ |   | ✓ | 72.2 | 67.6 | 73.6 | 56.3 | 49.7 | 58.7 |
| #4 | ✓ |   | ✓ | ✓ | 71.4 | 68.2 | 73.3 | 55.7 | 49.9 | 56.9 |
| #5 | ✓ | ✓ | ✓ | ✓ | 71.6 | 68.3 | 74.0 | 57.8 | 53.0 | 60.4 |

## 4.4 Ablation Study

In this subsection, we perform an in-depth analysis of CLIP2UDA with ablation studies on each component to highlight its strengths.

*4.4.1 Effectiveness of Different Components.* As shown in Tab. 2, we train five models for two scenarios, including #1 means that performing attribute encoder in the segmentation task by feeding depth and intensity maps; #2 and #3 mean performing VisPA in #1 and engaging in multi-modal interaction via C-Fus and S-Fus as the connection, respectively; #4 means that using both C-Fus and S-Fus without VisPA to implement multi-modal interaction; #5 combines all of our components.

Starting from #1, introducing an attribute encoder leads to a significant mIoU boost (69.4%→72.8%, 48.2%→ 56.4%). This highlights the robustness of the 3D attributes in MM-UDA. In addition, #2 and #3 outperform #1 by large margins, demonstrating that learnable prompts with visual cues can encourage the model to learn domain-invariant representations (increasing from 72.8%→73.2%/73.6%,

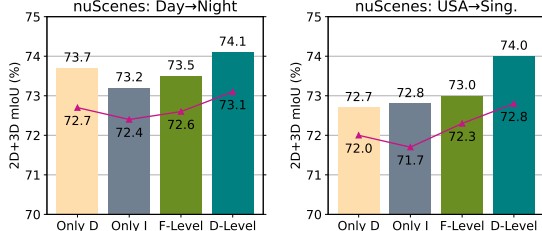

**Figure 5: Intra-modal combinations of 3D attributes.**

56.4%→58.1%/58.7%). Without VisPA, #5 still outperforms #2 by 0.8% and 0.5% in mIoU due to the densely integration of linguistic features and visual features. Ultimately, #6 combines all of our components to reach peak value.

*4.4.2 Intra-modal Combinations of 3D Attributes.* As shown in Fig. 5, we elaborate on the effectiveness of 3D attributes and their combinations (line graph), as well as the effectiveness of 3D attributes for multi-modal interaction under the guidance of the CLIP model (bar graph). Firstly, we investigate whether unified features constructed solely based on 3D attributes would enhance performance. Experimental results indicate that feeding 3D attributes into a 2D network enhances the robustness of distribution changes between source and target domains of multi-modality. Subsequently, to combine the visual features extracted from depth and intensity maps, we design two feature-level fusion strategies: 1) F-Level: Independent data inputs to two attribute encoders, followed by concatenation of the output features; 2) D-Level: Input the fused data into a single attribute encoder, resulting in the output of multi-attribute hybrid features. Compared to the former strategy, the

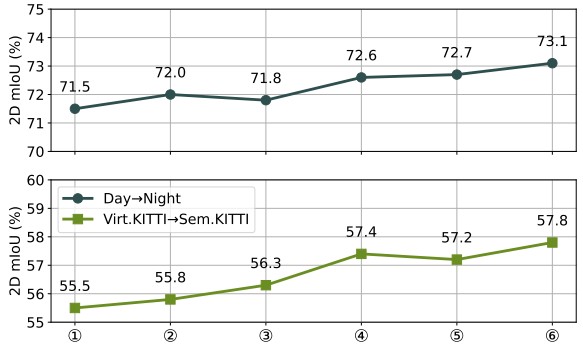

**Figure 6: Effect of visual cues for prompt design in 2D results.** ①: only I-token; ②: only D-token; ③: I-token + D-token; ④: I-token + $\mathcal{G}_T$; ⑤: D-token + $\mathcal{G}_T$; ⑥: I-token + D-token + $\mathcal{G}_T$.

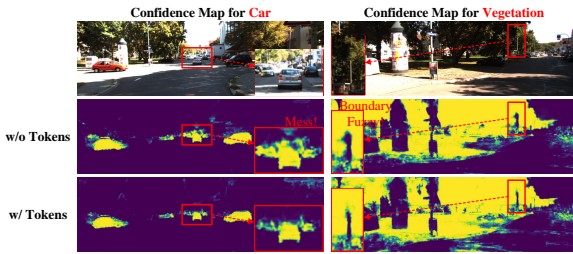

**Figure 7: Confidence maps on "Virt.KITTI→Sem.KITTI".**

improvement of "2D+3D" in the latter is remarked as less memory-consuming and performance-boosted, which obtains the mIoU gain of 0.5%/0.6% and 0.5%/1.0%, respectively. This phenomenon shows that while striving to maintain the strengths of each modality, it is also essential to consider the intra-modality correlation.

*4.4.3 Variety of Prompt Setting.* In this part, we illustrate the impact of prompt setting on the overall 2D performance of our method. As shown in Fig. 6, the results demonstrate a consistent upward trend, indicating that incorporating different learnable prompts to extract visual-driven textual embeddings improves performance. This observation suggests that the model benefits from incorporating visual information from multiple layers, enabling it to capture more nuanced and discriminative features. The reason we speculate is that the class prompts are independent of domains and are shared across all images and point clouds, while the image and domain prompts are independent of classes and represent specific visual information. In Fig. 7, we visualize confidence maps with and without appending prompts to demonstrate their effectiveness.

*4.4.4 Roles of CLIP.* In Tab. 3, we present results using the weights pre-trained CLIP model with fine-tuned and frozen strategy. In the fine-tuning stage, all models are utilized as 2D encoder $\mathcal{F}_I$. Among them, "Ours w/ finetune" achieves text-guided visual features in place of image features in the shared decoder. Compared to this setup, "Ours w/ frozen" (*i.e.*, CLIP2UDA) provides significant improvements of 1.1% mIoU on Day→Night and 1.5% mIoU on USA→Sing. We believe that our method inherently embeds local image semantics in its features as it learns to associate pixel and

**Table 3: Performance (mIoU, %) of different roles of CLIP in fine-tuned and frozen manner.**

| Role | Model | Day→Night | | | USA→Sing. | | |
|---|---|---|---|---|---|---|---|
| | | 2D | 3D | 2D+3D | 2D | 3D | 2D+3D |
| Finetune | CLIP-R50 | 69.2 | 71.0 | 71.2 | 67.4 | 65.1 | 69.6 |
| | DenseCLIP-R50 | 69.9 | 71.3 | 72.5 | 68.9 | 67.3 | 72.2 |
| | Ours-R50 | 71.0 | 71.4 | 73.0 | 69.8 | 67.5 | 72.5 |
| Frozen | Ours-R50 | 73.1 | 71.5 | 74.1 | 71.6 | 68.3 | 74.0 |
| | Ours-ViT-B-16 | 73.4 | 71.5 | 74.2 | 72.2 | 68.4 | 74.3 |

**Table 4: Performance (mIoU, %) of opposite domain adaptation on nuScenes.**

| Method | Night→Day | | | Sing.→USA | | |
|---|---|---|---|---|---|---|
| | 2D | 3D | 2D+3D | 2D | 3D | 2D+3D |
| Source-only | 55.1 | 70.3 | 64.7 | 62.2 | 68.4 | 71.3 |
| xMUDA | 67.4 | 71.1 | 71.9 | 69.2 | 70.0 | 73.5 |
| CLIP2UDA | 76.9 | 75.5 | 78.2 | 74.8 | 72.1 | 76.2 |
| $\triangle_{Gain}$ | ↑9.5 | ↑4.4 | ↑6.3 | ↑5.6 | ↑2.1 | ↑2.7 |
| xMUDA$_{PL}$ | 68.9 | 72.6 | 73.5 | 70.8 | 73.0 | 74.8 |
| CLIP2UDA$_{PL}$ | 78.4 | 76.0 | 79.7 | 75.4 | 72.7 | 76.8 |
| $\triangle_{Gain}$ | ↑9.5 | ↑3.4 | ↑6.2 | ↑4.6 | ↓0.3 | ↑2.0 |

point contents with natural language descriptions, indicating real benefit to feature transfer from language-compatible 2D features to 3D features.

*4.4.5 Opposite Adaptation.* To further test if our proposed method also works in the opposite adaptation direction, we run the experiments for "source-only", "xMUDA", "xMUDA$_{PL}$", and our methods "CLIP2UDA" and "CLIP2UDA$_{PL}$" on *nuScenes: Night→Day* and *nuScenes: Singapore→USA*. As shown in Tab. 4, the experimental results come as no surprise. Remarkably, compared to the 3D branch, CLIP2UDA effectively alleviates the performance degradation caused by 2D domain shift. Meanwhile, when comparing 2D+3D results our CLIP2UDA improves mIoU by +6.3%/+2.7% with respect to xMUDA, and +6.2%/+2.0% with extra pseudo-label supervision. This demonstrates that regardless of what target domain is given, CLIP2UDA can harness the frozen CLIP to uncover its pre-existing domain recognition capabilities.

## 5 Conclusion

In this work, we delve into the challenges faced by perceptual multi-models in the robustness of domain shifts in 3D scene understanding. To this end, we propose a top-performing method, dubbed CLIP2UDA, which makes frozen CLIP reward unsupervised domain adaptation in 3D semantic segmentation. Building upon CLIP, we introduce VisPA to leverage 2D feature response from the visual encoder to initiate learning of task-specific text prompts. Besides, we introduce TexCI to learn multi-modal domain-invariant representation in the shared decoder and obtain unified 2D visual predictions. We evaluate and discuss a comprehensive set of methods from related fields and study their adaptation across day-to-night, country-to-country, synthetic-to-real, and device-to-device datasets. Our findings underscore the need for more effective solutions in this region.

## Acknowledgments

This work was supported by the National Natural Science Foundation of China under Grant No.62176224, No.62176092, No.62222602, No.62306165; Natural Science Foundation of Chongqing under No.CSTB2023NSCQJQX0007; China Postdoctoral Science Foundation under No.2023M731957; China Computer Federation (CCF)-Lenovo Blue Ocean Research Fund; China Academy of Railway Sciences No.2023YJ357. Yuan Xie was supported by the CAAI-Huawei MindSpore Open Fund.

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
