# OpenReview forum: "CLIP2UDA: Making Frozen CLIP Reward Unsupervised Domain Adaptation in 3D Semantic Segmentation"
_acmmm.org/ACMMM/2024/Conference — MM2024 Poster_

### Official Review · Reviewer_fAzk · 2024-05-23

**Rating:** 4
**Confidence:** 3

**Summary:**

This paper presents CLIP2UDA, a method that leverages prompt learning to transfer the CLIP's generalization capability to MM-UDA. It preserves the pre-existing target information from the frozen CLIP model and learns vision-language-structure correlation.

**Strengths:**

The paper introduces CLIP2UDA, a novel method leveraging a frozen CLIP model for unsupervised domain adaptation, significantly enhancing 3D semantic segmentation performance.

Extensive Experimental Validation: The method demonstrates superior performance across various challenging scenarios, outperforming state-of-the-art techniques in multiple adaptation contexts.

**Limitations:**

1. The writing in the approach section needs improvement:

&emsp;	a. Due to the complexity of the method and the numerous modules, each module should be described from a more macro perspective. For instance, it is necessary to highlight the input of each module, the purpose of the transformation performed by the module, how the output is combined with other information, and how the modules/sub-modules collaborate. Additionally, Figure 2 should be more clearly organized, with the key modules proposed in this paper being prominently highlighted.

&emsp;	b. Given the numerous variables involved, they need to be appropriately labeled in the figure.

&emsp;	c. Why is C-Fus not shown in the figure? What is the relationship between C-Fus and S-Fus, and how are their results combined?

2. For I-token and D-token, can their attended information be presented through visualizations such as heatmaps?

**Suitability:**

3

---

### Official Review · Reviewer_AzQE · 2024-05-23

**Rating:** 2
**Confidence:** 3

**Summary:**

The paper presents Clip2UDA as a possible approach for utilizing CLIP in multi-modal domain adaptation for 3D semantic segmentation.

**Strengths:**

The task is interesting, while the idea of leveraging CLIP as a possible prior is intuitive, given CLIP's success in image tasks.

**Limitations:**

Although the idea is intuitive and interesting, the key concern for this paper is its lack of novelty, or more specifically, the lack of novelty for using CLIP for UDA.

Leveraging frozen CLIP (or denoted just as CLIP) as a prior knowledge for downstream task is not a new idea. Whether for image/video, or for 3D data. Just to list a few for reference:
1. Rozenberszki, D., Litany, O., & Dai, A. (2022, October). Language-grounded indoor 3d semantic segmentation in the wild. In European Conference on Computer Vision (pp. 125-141). Cham: Springer Nature Switzerland.
2. Chen, R., Liu, Y., Kong, L., Zhu, X., Ma, Y., Li, Y., ... & Wang, W. (2023). Clip2scene: Towards label-efficient 3d scene understanding by clip. In Proceedings of the IEEE/CVF Conference on Computer Vision and Pattern Recognition (pp. 7020-7030).
3. Zhou, Z., Lei, Y., Zhang, B., Liu, L., & Liu, Y. (2023). Zegclip: Towards adapting clip for zero-shot semantic segmentation. In Proceedings of the IEEE/CVF Conference on Computer Vision and Pattern Recognition (pp. 11175-11185).
4. Wang, Y., Huang, S., Gao, Y., Wang, Z., Wang, R., Sheng, K., ... & Liu, S. (2023, October). Transferring CLIP's Knowledge into Zero-Shot Point Cloud Semantic Segmentation. In Proceedings of the 31st ACM International Conference on Multimedia (pp. 3745-3754).

The idea of leveraging CLIP is not groundbreaking. And in this paper, it seems that CLIP is only leveraged to provide additional text information towards the RGB stream of xMUDA, from which the method is built upon. As such, the native constraint of CLIP, which is its incapability of dealing with non-visual data (more specifically non-RGB-based data) still exists. Adding in a good representation can indeed improve performances, yet it seems more of a engineering skill rather than a technical novelty. The authors would need to state specifically how CLIP is used differently and differently from other CLIP+UDA methods.

**Suitability:**

3

---

### Official Review · Reviewer_4RLd · 2024-05-23

**Rating:** 5
**Confidence:** 3

**Summary:**

This paper leverages CLIP to mitigate 3D performance degradation caused by 2D domain shifts, i.e, environmental changes. This paper proposes a novel method named CLIP2UDA, which learns task-specific prompt and multi-modal domain-invariant representations with the frozen CLIP model.

**Strengths:**

1. Utilizing CLIP in 3D Semantic Segmentation under domain shift is novel.
2. Clear and concise writing, well-presented figures, formulas, and tables.
3. Well-structured experiment setups with good analysis.
4. The performance improvement is significant.

**Limitations:**

1. In line 373, the example  "A yellow car driving in the night scene.” is more exact than "A car in the scene."  may not be suitable. Because prompts [V] is a learnable vector that may not have a specific meaning.
2. The author should briefly introduce the comparison method in quantitative and qualitative comparison.
3. How about the results on the ViT-based CLIP?

Despite the limitations that should be addressed, I don't believe they should impact the paper's consideration for publication.

**Suitability:**

3

---

### Meta-Review · Area_Chair_czfC · 2024-07-02

**Recommendation:** Accept (Poster)
**Confidence:** 5

**Metareview:**

Pros:
- Utilizing CLIP in for domain adaptive 3D Semantic Segmentation is novel.
- The performance is good.
- The paper is well-written and easy to understand.

Cons:
- The paper lacks comparisons to other methods that use CLIP for UDA.
- Some figures need to be improved.